# Liquid Biopsy as a Tool for the Characterisation and Early Detection of the Field Cancerization Effect in Patients with Oral Cavity Carcinoma

**DOI:** 10.3390/biomedicines9101478

**Published:** 2021-10-15

**Authors:** Elisabeth Pérez-Ruiz, Vanesa Gutiérrez, Marta Muñoz, Javier Oliver, Marta Sánchez, Laura Gálvez-Carvajal, Antonio Rueda-Domínguez, Isabel Barragán

**Affiliations:** 1Unidad de Gestión Clínica Intercentros de Oncología Médica, Oncology Department, Institute of Biomedical Investigation of Malaga (IBIMA), Hospitales Universitarios Regional y Virgen de la Victoria, 29010 Malaga, Spain; vanesa_gutierrez78@hotmail.com (V.G.); ayllon.m@gmail.com (M.M.); lauragalvezcarvajal@hotmail.com (L.G.-C.); 2Researcher Unit, Unidad de Gestión Clínica Intercentros de Oncología Médica, Institute of Biomedical Investigation of Malaga (IBIMA), Hospitales Universitarios Regional y Virgen de la Victoria, 29010 Malaga, Spain; javiom@gmail.com (J.O.); or isabel.barragan@ki.se (I.B.); 3Maxillofacial Surgery Department, Hospital Regional Universitario de Málaga, 29010 Málaga, Spain; sanchezmarta11q@gmail.com; 4Group of Pharmacoepigenetics, Department of Physiology and Pharmacology, Karolinska Institutet, 171 77 Stockholm, Sweden

**Keywords:** oral cell carcinoma, field cancerization, liquid biopsy

## Abstract

Oral squamous cell carcinoma (OSCC) constitutes approximately 25% of all head and neck cancer, for which the consumption of tobacco and alcohol are the main associated risk factors. The field cancerization effect of OSCC is one of the main reasons for the poor survival rates associated with this disease. Despite some advances, its ccharacterization and early diagnosis continue to challenge modern oncology, and the goal of improving the prognosis remains to be achieved. Among new early diagnostic tools for OSCC that have been proposed, liquid biopsy appears to be an ideal candidate, as studies have shown that the analysis of blood and saliva provides promising data for the early detection of relapses or second tumours.

## 1. Oral Cavity Carcinoma

Head and neck squamous cell carcinoma (HNSCC) is the sixth most common neoplasm worldwide [1,2]. Within this group of tumours, oral squamous cell carcinoma (OSCC) constitutes approximately 25% of all cases, for which the consumption of tobacco and alcohol are the main associated risk factors. Despite the ease of self-examination, this condition is usually diagnosed at locally advanced stages when regional lymph nodes have been affected. The main treatment option is surgical resection, combined with adjuvant radiotherapy or radiochemotherapy in patients at high risk of relapse [3,4]. Although multidisciplinary treatment is readily available, advanced OSCC has a poor prognosis, and only half of these patients will be disease-free at five years after surgery [5]. Moreover, this rate has not improved in the last decade despite the intensification of adjuvant treatments and preoperative chemotherapy [6], fundamentally due to a central biological aspect of this pathology: the field cancerization effect [7].

## 2. Field Cancerization Effect in OSCC: Pathophysiology and Associated Problems

The concept of field cancerization suggests that, in certain persons, the mucous membranes of the upper aerodigestive tract, when subjected to common carcinogens, are at greater risk of developing new carcinomas. Since this term was coined by Slaughter et al. in 1953 [7], it has been used to describe various types of pre-malignant disease, when a higher than expected prevalence of multiple local primary or secondary tumours is observed, accompanied by synchronous distant tumours. Slaughter studied the OSCCs obtained from 783 patients and detected multiple independent tumours in 11.2%. 

The monoclonal theory proposed to explain this field effect suggests that a single cell or a small group of cells (cluster) undergoes a malignant transformation, and that it is its dissemination (via saliva or intra-epithelial migration) that causes the appearance of multiple tumours. In contrast, the polyclonal theory is based on the premise that any transformation event is rare and that the existence of multiple lesions arises from the generalised migration of transformed cells throughout the aerodigestive tract [8,9].

In recent years, molecular techniques have made it possible to extend our understanding of the relationships among these lesions, although the field remains controversial [10]. One of the challenges in OSCC is to define the degree of cancerization of a field effect, which involves differentiating between a mutant lineage (the presence of multiple mutations which, however, do not provoke carcinogenesis) and a cancerized lineage (characterized by mutations that in a suitable microenvironment result in tumour development) [11].

As observed above, the multifactorial aetiology and the characteristic field cancerization effect of this neoplasm are responsible for the appearance of multiple tumours in the oral cavity, the definitive diagnosis of which is made by solid biopsy of the tissue affected. The latter procedure, however, when conducted in an area that has been previously treated for the presence of a tumour, can generate significant comorbidity, is expensive and remains invasive. Furthermore, the solid biopsy approach has three important limitations: (1) it does not characterise the tumour heterogeneity in the tissue; (2) it may obtain insufficient material, or the suspicious injury may be in a dangerous area for re-biopsy, which would hamper or prevent complementary molecular study; and (3) the molecular profile of a relapse or progression of the disease might differ from that of the primary tumour, in which case it would be necessary to perform a re-biopsy, which for some patients would not be possible [12].

These considerations help explain why oral cavity neoplasms, their recurrences and/or second primary tumours are usually diagnosed late, which worsens the patient’s prognosis. Therefore, to improve survival for patients with OSCC, the early detection and diagnosis of relapse and second tumour is vital. 

### 2.1. Liquid Biopsy

In this respect, liquid biopsy has been proposed as a simple and effective tool, which does not generate comorbidity, is minimally invasive and provides valuable help in managing OSCC. This technique is based on the detection of tumour-related components in the bloodstream, saliva, cerebrospinal fluid or any other body fluid, and can identify circulating tumour cells (CTC) and circulating genetic material such as genomic DNA (gDNA), mitochondrial DNA (mtDNA), circulating tumour DNA (ctDNA), microRNA (miRNA) or exosomes. 

Liquid biopsy is not only a highly accessible technique, it is also the ideal tool for the personalised study of tumour evolution, since it allows us to analyse the genetic variations that occur during the disease process, thus facilitating early diagnosis (as has been demonstrated with respect to other neoplasms) [13,14,15,16]. In the next sections, we will try to justify the potential role of liquid biopsy as a tool to characterize the effect of field cancerization in the OSCC. 

### 2.2. ctDNA in Oral Cavity Tumours 

Physiological and pathological processes in the body cause different forms of cell death and determine the types of material deposited in the body fluids. The determination of circulating cell-free DNA (cfDNA) in humans was the first step towards enabling liquid biopsy [17]. Subsequently, the detection of specific molecular aberrations of each tumour within the cfDNA made it possible to characterize a tumour-specific fraction of circulating DNA, termed the circulating tumour DNA (ctDNA) [18,19,20] which represents less than 1% of cfDNA and has significant potential as a biomarker in oncology [21,22]. The detection of ctDNA in cancer patients after treatment with curative intent indicates the presence of minimal residual disease, and is a prognostic marker of relapse for various tumours [23,24]. Most ctDNA studies of head and neck tumours have been carried out as part of broader investigations involving different types of tumours, such as breast, ovary, prostate, colorectal and lung. Lawrence MS et al. [24] describes the molecular profile of HNSCC, reporting *TP53* (72%), *PIK3CA* (21%), *FAT1* (23%), and *CDK2NA* (22%) as key mutations in these tumours. Lebofsky et al. [25] studied the concordance between ctDNA mutations and those observed in histological samples, and found that three patients with HNSCC presented 97% concordance between the ctDNA and the biopsies. The same study also found that the higher the tumour load, the higher the plasma concentration of ctDNA.

Many studies have been undertaken to consider the role of ctDNA in tumours of the upper aerodigestive tract. However, those focusing on the use of liquid biopsy in OSCC are few and heterogeneous. Table 1 summarizes the main studies of this question.

Among the studies published in the present century, of particular interest for our purposes are those by Nunes et al., Hamana et al. and Kakimoto et al. [26,27,28], all of which considered the level of microsatellite instability as a marker in patients with OSCC. Nunes et al. [26] analysed eight markers in tissue and serum and detected this alteration in 58% of cases (regardless of stage). Moreover, 17 patients presented the same alteration in plasma, leading the authors to conclude that this analysis could be used as a means of early diagnosis of OSCC. In contrast, Hamana et al. [27] and Kakimoto et al. [28], who each analysed nine markers of instability in the blood and tissue of patients before and after surgery, observed that those with allelic imbalance after surgery underwent relapse, a finding that suggests early biomarkers of OSCC relapse might be achieved. However, these studies were based on small samples of patients and their results have not been validated in prospective studies.

Shukla et al. [29] studied 150 patients with OSCC, using spectrometry to measure the quantity of cfDNA in plasma. These values were then compared with the cfDNA in 150 post-surgical OSCCs and in 90 pre-malignant lesions. No significant differences were observed between the groups of patients, and so the authors concluded that this analysis did not contribute to the early identification of pre-malignant lesions.

In an alternative approach, Wang et al. [30] explored the usefulness of ctDNA as a biomarker in plasma and saliva in 93 patients with HNSCC at different stages and in different anatomical locations. The results obtained showed that when a joint evaluation of plasma and saliva is performed, ctDNA is detected in >90% of the patients. Moreover, the authors observed that sensitivity for the detection of ctDNA in saliva was dependent on the location of the primary tumour (the test was more effective in tumours of the oral cavity) and on the disease stage. In early stages, saliva appears to be a more sensitive predictor than plasma (100% vs. 70% respectively), while the latter is more sensitive in advanced stages (92% vs. 70% in saliva). In a related investigation, Mazurek et al. [31] analysed 200 patients with HNSCC, of whom 78 had oropharyngeal carcinoma, while the rest were non-oropharyngeal. This study detected higher concentrations of ctDNA in tumours of the oropharynx and in patients with a higher disease stage. Overall, 14% of these patients were Human Papilloma Virus (HPV)+. The virus was detected in 86% of plasma samples and 40% of saliva samples from the patients affected, while *EGFR* and *KRAS* were not detected in any case. The authors concluded that HPV cfDNA tests might facilitate early diagnosis.

More recently Perdomo et al. [32] used two approaches—targeted sequencing in five genes, and the sequencing of the entire coding region of the *TP53* gene—to detect mutations in ctDNA and to determine the level of detection concordance between fluids and tissue in a cohort of patients that included 41 with OSCC. The results obtained demonstrated the potential value of detecting targeted mutations in the ctDNA of patients in early stages of HNSCC. However, the concordance in saliva was low; indeed, the analysis of *TP53* mutations in healthy individuals revealed the presence of pathogenic mutations, which should be taken into account in any subsequent design of ctDNA assays for early diagnosis.

Finally, a recent study by Shanmugam et al. [33] used massive sequencing techniques to analyse tissue and saliva samples from 121 pre-surgical patients with OSCC. Alterations were identified in more than 95% of these patients, and in more than 97% the alteration was present in saliva, although this concordance was lower in initial disease stages. The authors concluded that new massive sequencing techniques could enhance screening and the early detection of relapses.

In summary, the study of ctDNA, using either simple digital polymerase chain reaction (PCR) techniques or more complex ones based on next-generation sequencing, detects genetic alterations in the primary tumour, which can then be monitored and observed. However, as commented above, OSCCs are characterized by high levels of recurrence due to the field cancerization effect. As a result, performing peripheral blood monitoring (ctDNA) for molecular alterations in the primary tissue can obtain false negatives during follow-up if additional primary neoplasms appear. Therefore, further determinations are required, whether conducted singly or in conjunction with the determination and/or sequencing of ctDNA, in order to offset the field effect of the OSCC.

### 2.3. Exosomal RNA

As the cfDNA, miRNAs, either free in plasma or included in extracellular vesicles called exosomes, have been extensively studied in recent years. miRNAs are small non-coding RNAs with approximately 22 nucleotides that are responsible for regulating the expression of various genes at the post-transcriptional level. The expression of miRNA is altered in malignant tumours [34]. 

Many studies have been undertaken to clarify the role of miRNAs in OSCC. Although the use of different methodologies limits the reproducibility of the results obtained [35], some miRNAs associated with the diagnosis of OSCC have been described, and variations in their levels of expression have been described after surgery for this condition. Reported findings include the overexpression of *mir21* and *mir31* and the downregulation of *mir200a* and *mir125a* [36] in the blood of patients with OSCC, in comparison with healthy subjects. A recent review describes the usefulness of these small RNAs as biomarkers for diagnosis; moreover, in vitro studies suggest they can also be used as therapeutic targets [37]. Furthermore, many of the miRNAs described are detected in saliva, which highlights the importance and accessibility of this fluid in the management of OSCC tumours [38,39,40,41,42]. A recent meta-analysis demonstrates the usefulness of determining the presence of miRNAs in blood or saliva as a tool for the early diagnosis of OSCC [35].

The extracellular vesicle population is comprised of different types of vesicles, according to their structure and content. These include microvesicles, exosomes and apoptotic bodies, each performing specific functions [43]. Many cell types, such as dendritic cells, B cells, T cells, mast cells, epithelial cells and tumour cells, can produce exosomes [44]. Unlike other microvesicles, exosomes do not initially bear nuclear DNA, but may contain coding and non-coding mitochondrial DNA, proteins, peptides, lipids and nucleic acids (messenger RNA, miRNA, circular RNA), with biological activity [45].

Exosomes are involved in different stages of tumour development, including carcinogenesis, growth and development (by transferring different types of molecules, exosomes modulate pathways and regulate gene activation). In addition, they play a significant role in angiogenesis, in neutralising the immune response (by transmitting suppressor signals to cells in the immune system or by inhibiting the activation of receptors of these cells) [46] in developing resistance to chemotherapeutic agents and, finally, in metastasis [47,48]. 

Due to their unique biogenesis and their ability to circulate freely throughout the body, exosomes have been proposed as potential biomarkers in a liquid biopsy [49]. Many studies have shown that the morphology and content of exosomes isolated from the saliva and blood of patients with OSCC differs greatly from those of healthy people, suggesting that exosomes might be used to diagnose OSCC at an early stage [50]. Furthermore, OSCC is composed of different cell types, each of which can secrete exosomes containing a unique set of miRNAs. For example, it has been shown that *miR-200c-3p* is capable of inducing invasive potential in non-invasive cells within a tumour mass in patients with OSCC [51]. To date, only a small fraction of the RNA contained in exosomes has been identified [52]. The miRNA described can play various roles in the development of OSCC, during growth (*miR-142-3p*), migration and invasion (*miR-200c-3p, miR3825p, miR-21*), and in the development of metastases (*miR-21-5p, miR-34a-5p, miR-29a-3p*) [39,51,53,54].

Exosomes secreted by tumour cells are involved in angiogenesis and the development of metastasis under hypoxic conditions [48]. A miRNA expression profile of exosomes derived from hypoxic tumours revealed higher levels of *miR-21, miR-205* and *miR-148b* compared to exosomes examined at a normal oxygen concentration. Furthermore, circulating levels of exosomal *miRNA-21* were associated with the expression of *HIF-1α/HIF-2α*, T-stage and lymph node metastasis in patients with OSCC. These findings suggest that a hypoxic microenvironment may promote a prometastatic reaction, stimulating tumour cells to generate *miRNA-21*-rich exosomes that are delivered to normoxic cells [39]. As *miR-21*, high levels of *miR-483-5p* in serum from patients with oral cancer was correlated with tumour stage and lymph node metastasis [55,56].

Recently, several studies have found that combining several miRNA, the specificity for the diagnosis of OSCC is significantly increased, highlighting *miR-370-3p* and *miR-30a-5p* [57]. Other studies have shown that the plasma levels of *miR-375-3p*, *miR-138-5p* and *miRNA-99-5p* were associated with the clinical development of patients with OSCC [58]. 

In other studies, elevated levels of *miR-24* in plasma have been associated with high sensitivity (AUC 0.82) in OSCC patients and with clinical stage. In vitro studies have been reported that the mechanism of action is through interaction with *FBXW7* [59,60].

Tachibana et al. (2016) were able to identify 20 miRNA differentially expressed in tissue and plasma samples from patients with gingival squamous tumours using microarrays. It should be noted that *miR-233* was found to be over-expressed in plasma but down-regulated in the tissue, suggesting that it could be a local defence mechanism to inhibit local tumour growth [61].

### 2.4. Exosomes in Blood

Along with miRNAs, exosomes can also contain proteins, both inside and in their membranes. In the case of OSCC, some of the proteins contained in exosomes have been analysed, observing that they can promote tumorigenesis and regulate stromal cells.

An example is the secretion of exosomes containing epithelial growth factor receptor (*EGFR*), which can be heightened by the stimulation of its ligand (EGF). When vesicles are internalised by healthy epithelial cells around cancer cells, this produces an epithelium-mesenchyme transition, by which normal cells are converted into spindle cells, thus promoting cell invasion and migration within the tumour medium [62]. 

A study analysed the exosomal surface proteins CD63 + and CAV1 + before and after surgery, in OSCC patients, and it was observed that low levels of CAV1 + exosomes before surgery were related to longer survival. Furthermore, a significantly lower expression of CD63 + exosomes was observed after surgery. This result seems to suggest that the tumour mass is responsible for the high levels of circulating exosomes detected in patients with this tumour type [56]. Therefore, the authors suggest that the monitoring of CD63 + exosome levels after tumour resection can be very useful for the clinical follow-up of patients with OSCC [56].

The use of immune checkpoint inhibitors of the PD1/PD-L1 pathway was a major advance in the treatment of recurrent and metastatic squamous cell carcinomas of the head and neck [63]. Exosomes can transport PD-L1, which affects the modulation of the tumour microenvironment and the disease progression. In this respect, Theodoraki et al. isolated exosomes from the plasma of 40 patients with HNSCC, observing that these exosomes carried biologically active PD-L1, which was correlated with the tumour stage and with lymph node involvement [64]. 

### 2.5. Exosomes in Saliva

The determination of exosomes in saliva is simple and non-invasive. Saliva contains less protein than blood, which greatly simplifies the identification and quantification of exosomes. Moreover, it can be stored at 4 ° C (without the need for freezing at −80 °C), which facilitates storage and conservation [65].

Recently, a signature of miRNA from saliva exosomes of OSCC patients has been reported, and it was observed that the miRNA *miR-320-3p* and *miR-517b-3p* and *miR-320-3p* were significantly represented and specified with an AUC of 0.87 [66]. 

Previously, massive expression studies in direct saliva samples (without isolating exosomes) that included patients and healthy controls showed that several miRNAs have been found with low expression, including *miR-125a*, *miR-147*, *miR-136, miR-148a, miR-200, miR-323-5p, miR-668 miR-503, miR-632, miR-646, miR-877*, and *miR-1250*; on the other hand, two miRNAs (*miR-24* and *miR- 27b*) were up-regulated [67,68]. 

In 2011, two types of salivary exosomes were described, differing mainly in their size and protein composition. Type I exosomes are larger and denser to electrons, while type II are more similar to those present in other body fluids [69]. Salivary exosomes participate in the catabolism of biopeptides and play an important role in the local immune response of the oral cavity. 

In 2016, an in vitro study by Ogawa et al. showed that the content of salivary exosomes could be transferred horizontally to other cells, modulating the gene expression of the receptors and increasing their invasive and migratory capacity [70].

Langevin et al. used mass sequencing to differentiate the miRNAs from tumour cells from those of healthy epithelial cells. These authors observed that many miRNAs were common to the exosomes of both healthy and cancerous cells. However, significantly higher concentrations of *miR-486-5p, miR-486-3p* and *miR-10b-5p* were detected in the tumour cells than in the control group. This finding provided important information on tumour biology and generated a novel set of biomarkers to discriminate between OSCC and potentially malignant oral lesions, which is crucially important to the goal of preventing the field effect of these tumours [71].

Finally, Zlotogorski-Hurvitz et al. detected morphological and molecular differences in the salivary exosomes of patients with OSCC with respect to healthy patients, and suggested that this feature could be employed as a screening measure for high-risk patients. These authors analysed the expression of three exosomal markers (CD9, CD81 and CD63) and found a higher mean concentration of CD63 exosomes and a lower one of CD9 and CD81 exosomes between cancer patients and healthy individuals, although significant statistical differences were only observed with respect to CD81 exosomes [50].

### 2.6. Circulating Tumour Cells (CTCs)

CTCs are released by the primary tumour or by metastatic lesions into the bloodstream, and therefore share most of their mutational profile with that of the tumour clones present in the primary tumour. CTCs may circulate alone or in groups, the latter having greater metastatic potential [72]. One of the characteristics of CTCs is their low ratio, approximately 1 CTC per 10^7^ white cells per millilitre of blood in metastatic patients. Various techniques can be used to determine the quantity of CTCs present, but the CellSearch^®^ platform is the only one approved for prognostic use in patients with breast, prostate or colorectal cancer [73]. 

Most studies of OSCC have focused on validating CTCs as prognostic biomarkers and predictors of relapse by reference to CTC levels or their genomic characterisation [74,75]. 

A 2014 study reported data on the prognostic value of the CTC count in 110 patients with different stages of OSCC, the correlation of this value with clinical and pathological parameters, and rates of relapse or death. The authors also analysed the presence of disseminated tumour cells (DTCs) in the bone marrow of the iliac crest. The recorded presence of CTCs was 12.5% with a range of 1-14 CTCs/7.5 mL. Detection was significantly correlated with tumour size and with the presence of distal metastases, but no statistically significant association was found with overall survival or with time to relapse or death, probably due to the small sample size [76]. Although no significant correlation was found between the detection of CTCs and DTCs, their joint presence was significantly associated with relapse-free survival time. 

A later study analysed the prognostic value of CTCs in 40 patients with OSCC, either locally advanced (38%) or oropharyngeal (63%). All patients were treated with induction therapy, with docetaxel, cisplatin and fluorouracil, surgery, and postoperative adjuvant radiation therapy. At baseline, CTCs were detected in 80% of the patients; at the completion of treatment, 3% had no CTCs. Higher CTC levels were associated with a higher risk of recurrence during treatment and with lower overall survival. Postsurgical CTC levels were elevated (although they decreased after radiotherapy), but were not associated with a negative prognostic impact [74].

In addition to the CTC count, the molecular characterisation of CTCs in OSCCs has been addressed using EGFR, podoplanin or EpCAM, among other parameters. In this respect, Oliveira-Costa et al. [77] analysed the gene expression profile of a series of oral cavity tumours, seeking to identify biomarkers by searching for genes whose expression is increased or decreased during their natural history. A total of 879 transcripts increased or decreased significantly from stage T1 to T4. Of these, six (*PD-L1, HOXB9, DHDH, BLNK, ZNF813* and *IL6ST*) were validated by qRT-PCR in tissue samples. Consideration of four of these markers in CTCs revealed an increase in the expression of *PD-L1, HOXB9* and *ZNF813* and a significant decrease in that of BLNK. In view of these findings, the authors suggested that patients with CTC PD-L1+ might benefit from anti-PD-L1 therapy. 

Strati et al. conducted a prospective study of a cohort of patients with locally advanced carcinoma of the head and neck, and reported that the evaluation of PD-L1-overexpressing CTCs in liquid biopsies is feasible and may provide significant prognostic information. These considerations have significant implications for patients being treated with PD1 inhibitors [78]. 

CTC analysis is a developing technique in the field of HNSCC in general and in that of OSCC in particular. However, a factor of crucial importance in any future clinical implementation is the need to improve the sensitivity of the techniques used to quantify and characterise the presence of CTCs.

In Table 2, we summarise the potential role of different liquid biopsy technical approaches for the diagnosis and prognosis of OSCC.

## 3. Conclusions

The field cancerization effect of OSCC is one of the main reasons for the poor survival rates associated with this disease. Despite some advances, its characterisation and early diagnosis continue to challenge modern oncology, and the goal of improving the prognosis remains to be achieved.

Among new early diagnostic tools for OSCC that have been proposed, liquid biopsy appears to be an ideal candidate, as studies have shown that the analysis of blood and saliva provides promising data for the early detection of relapses or second tumours.

Determining the presence of ctDNA or of specific mutations of primary tumours via the analysis of blood or saliva may play a useful role in screening high-risk patients, especially if this analysis is conducted via massive sequencing techniques enabling a broad-based characterisation. A good example of such utility that has been adopted to clinical practice is the assessment and tracking of cfHPV DNA in HPV related oropharyngeal cancer, both at diagnosis and as treatment efficacy monitoring. However, given the possibility of further primary tumours or of field effect tumours with molecular alterations differing from those of the primary tumour, it remains necessary to study other liquid biopsy components in order to enhance the characterisation of tumourigenesis in patients with OSCC.

The study of CTCs remains open and the reliable early detection of tumour recurrence is a goal that has yet to be met, as the sensitivity and reproducibility of the techniques described remain far from optimal. Nevertheless, we believe CTCs could play a significant role in characterising recurrent/metastatic OSCC, as has been demonstrated for other tumours.

Finally, the study of free miRNAs, whether circulating or incorporated in exosomes, as well as the study and characterisation of exosomes themselves, have shown promising results. Although the results of different studies have not been replicated, recent improvements in sequencing have facilitated massive data analyses that have yielded a wealth of new information on the role of miRNAs and exosomes in OSCCs.

## Figures and Tables

**Table 1 biomedicines-09-01478-t001:** Studies of cell free DNA (cfDNA)/circulating tumour DNA (ctDNA) in blood and/or saliva samples from oral squamous cell carcinomas (OSCCs).

Authors, Year	OSCCs (*n*)	Other Tissue (*n*)	Technique/Detection	Results	Discussion/Conclusion
Nunes et al. (2001) [26]	46	45	Eight microsatellite markers in tissue and serum (cfDNA)	58% had microsatellite alterations and 17 patients had the same profile in plasma.	Early detection.
Hamana et al. (2005) [27]	64	None	Nine microsatellite markers in tissue and serum (pre, immediately post-surgery and 4 weeks after surgery	Allelic imbalance patterns in serum were associated with the presence of allelic imbalance in paired tumour tissue. Patients with allelic imbalance four weeks after surgery developed metastases	Microsatellite analysis could help assess the risk of recurrence.
Kakimoto et al. (2008) [28]	20	None	Nine microsatellite markers in tissue and serum (one month before and after surgery)	Allelic imbalance in ctDNA was observed in blood in 90% of patients	Microsatellite analysis could help assess the risk of recurrence.
Shukla et al. (2013) [29]	150 OSCCs150 post-treatment OSCCs	90 potentially malignant lesions	Quantity of cfDNA in plasma by spectrophotometry	No differences	Rich lymphatic drainage of the oral mucosa prevents it from entering the bloodstream.
Wang et al. (2015) [30]	15	78	Pre and post-treatment (9) samples of blood and saliva to detect ctDNA and mutations by multiplex PCR.	- ctDNA in blood and saliva was found in 96% of 47 patients. - ctDNA was present in 100% of saliva and blood specimens of OSCCs. - After treatment, four patients with ctDNA underwent recurrence.	Utility in monitoring. Saliva provides a more sensitive predictor of early-stage disease than plasma
Mazurek et al. (2016) [31]	Unknown	200	*HPV16/18*, *KRAS* and *EGFR* by q-PCR	14% were HPV16+ Neither *EGFR* and *KRAS* were detected. *HPV* was found in 86% of plasma and 40% of saliva specimens.	*HPV* cfDNA could be used for the early detection and monitoring of HPV+.
Perdomo et al. (2017) [32]	41	Other head and neck cancer	Approaches: (a) Mutations in ctDNA of five genes identified in tissue and plasma (b) *TP53* mutation analysed in tissue, plasma and oral rinses.	(a) 18 mutations in 42% of patients (b) 36%, 3% and 26% of *TP53* mutation (tissue, plasma and oral rinses)	Concordance of mutation detection was low between tumour tissue, oral rinses and plasma.
Shanmugam et al. (2021) [33]	121	None	Next generation sequence (NGS) with tissue and saliva specimens (before surgery)	In 95.87% of cases, at least one somatic variant was identified. The most prevalent mutated genes were *TP53*, *FAT1*, *CDKN2A* and *NOTCH1* (the same as in tumour tissue). Concordance between mutations in tumour and saliva was >97% but less in early-stage disease.	Sequencing platforms could be used to screen high-risk individuals, facilitating early detection and monitoring.

**Table 2 biomedicines-09-01478-t002:** Liquid biopsy technical approaches for oral squamous cell carcinoma (OSCC) diagnosis and prognosis.

Liquid Biopsy Techniques	Diagnosis/Prognosis	References
ctDNA	Diagnosis > Prognosis	[26,32,33]
miRNA	Diagnosis > prognosis	[35,36]
Exosomes	Diagnosis	[50,71]
CTC	Prognosis	[74,75]

ctDNA: circulating tumoral DNA; miRNA: microRNA; CTC: circulating tumours cells.

## Data Availability

Not applicable.

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
