# Peer review of "Liquid Biopsy as a Tool for the Characterisation and Early Detection of the Field Cancerization Effect in Patients with Oral Cavity Carcinoma"

_biomedicines, 2021, doi:10.3390/biomedicines9101478_

Round 1
Reviewer 1 Report
Comments to the Author
Pérez-Ruiz and co-workers present a study that propose potential role of liquid biopsy as a tool to characterize the effect of field cancerization in the OSCC (Oral squamous cell carcinoma). They try to analyse the genetic variations that occur during the disease process, thus facilitating early diagnosis.
They conclude that liquid biopsy appears to be an ideal candidate, as studies have shown in the analysis of blood and saliva and can provide promising data for the early detection of relapses or second tumours.
Generally the technical part of this work seems to be well conducted and performed. The procedures and techniques used are standard and appear appropriate.
I believe that gathering and joining all the information known to date on this subject is very useful for having an overview of the situation and to direct and update the clinician in the treatment OSCC patients. Moreover this study seems to me very useful for an early OSCC characterisation and diagnosis.
Author Response
Dear Reviewer,
Thank so much for your comments. We have started our study and we hope to obtain good results in the next years. It will be a great news for our patients.
Please, if you have interesting in our study and you want colaborate, please write me an e-mail.
Best regards,
Elisabeth
Reviewer 2 Report
This manuscript focused on the new diagnostic tool, liquid biopsy, in oral squamous cell carcinoma (OSCC). In this article, the authors discussed several different tumor-related components from blood, saliva, and cancer tissues, such as circulating tumor cells, circulating tumor DNA, and microRNA in patients with OSCC. The manuscript is well organized with the impressive application.
However, there are some minor errors in this manuscript:
- Please use the unity term in the whole context, for example, cancerization and cancerisation (Line 22, 45, 67, 171).
- The abbreviation of cfDNA and ctDNA is confusing for the reader.
- Please use the unity abbreviation as miRNA (Line 196 and 251).
- “ Oral cancer “ is better to replace by OSCC in the whole manuscript. (Line 240 and 243)
- Please correct the typing error miR-370-3p y miR-30a.(Line 243)
- Add the subtitle “Exosomes in blood” since the authors discussed Exosomes in saliva.
- Add a simple table to list all the techniques of liquid biopsy and the predicted application (diagnosis or prognosis) in OSCC.
Author Response
Please, review the attached file.

Reviewer 3 Report
Dear Authors,
The manuscript is well written and describe in comprehensive way the possibility of liquid biopsy in early tumor detection and monitoring patients after treatment. In my opinion later one is more clinically important because about 50% will relapse after primary treatment and early detection of such failure may enable effective salvage. This utility of liquid biopsy with examples from literature should by more stressed and discussed in the manuscript. The good example of such utility that has been adopted to clinical practice is the assessment and tracking cfHPV DNA in HPV related oropharyngeal cancer both at diagnosis, as treatment efficacy monitoring , in early assessment of treatment results and in follow-up for early detection of tumor relapse prior to radiological or clinical symptoms of recurrence. This fact should be also mentioned in the manuscript.
Author Response
Please, review the attached file.
